# Miltefosine: A Repurposing Drug against Mucorales Pathogens

**DOI:** 10.3390/jof9121166

**Published:** 2023-12-04

**Authors:** Mariana Ingrid Dutra da Silva Xisto, Rodrigo Rollin-Pinheiro, Victor Pereira Rochetti, Yuri de Castro-Almeida, Luana Pereira Borba-Santos, Giulia Maria Pires dos Santos-Freitas, Jefferson Cypriano, Fernanda de Ávila Abreu, Sonia Rozental, Eliana Barreto-Bergter

**Affiliations:** 1Laboratório de Química Biológica de Microrganismos, Departamento de Microbiologia Geral, Instituto de Microbiologia Paulo de Góes, Universidade Federal do Rio de Janeiro, Rio de Janeiro 21941-902, Brazil; rodrigorollin@gmail.com (R.R.-P.); victorrochetti@gmail.com (V.P.R.); yuricastro20155@gmail.com (Y.d.C.-A.); giuliapires89@gmail.com (G.M.P.d.S.-F.); 2Laboratório de Biologia Celular de Fungos, Programa de Biologia Celular e Parasitologia, Instituto de Biofísica Carlos Chagas Filho, Universidade Federal do Rio de Janeiro, Rio de Janeiro 21941-902, Brazil; luanaborba@biof.ufrj.br (L.P.B.-S.); rozental@biof.ufrj.br (S.R.); 3Laboratório de Biologia Celular e Magnetotaxia & Unidade de Microscopia Multiusuário, Instituto de Microbiologia Paulo de Góes, Universidade Federal do Rio de Janeiro, Rio de Janeiro 21941-902, Brazil; jeffcy@micro.ufrj.br (J.C.); fernandaaabreu@micro.ufrj.br (F.d.Á.A.)

**Keywords:** miltefosine, Mucorales, *Rhizopus oryzae*, *Rhizopus stolonifer*, *Rhizopus microsporus*, *Cunninghamella* spp., *Mucor velutinosus*, drug repurposing

## Abstract

Mucorales are a group of non-septated filamentous fungi widely distributed in nature, frequently associated with human infections, and are intrinsically resistant to many antifungal drugs. For these reasons, there is an urgent need to improve the clinical management of mucormycosis. Miltefosine, which is a phospholipid analogue of alkylphosphocholine, has been considered a promising repurposing drug to be used to treat fungal infections. In the present study, miltefosine displayed antifungal activity against a variety of Mucorales species, and it was also active against biofilms formed by these fungi. Treatment with miltefosine revealed modifications of cell wall components, neutral lipids, mitochondrial membrane potential, cell morphology, and the induction of oxidative stress. Treated Mucorales cells also presented an increased susceptibility to SDS. Purified ergosterol and glucosylceramide added to the culture medium increased miltefosine MIC, suggesting its interaction with fungal lipids. These data contribute to elucidating the effect of a promising drug repurposed to act against some relevant fungal pathogens that significantly impact public health.

## 1. Introduction

Mucorales are a group of non-septated filamentous fungi widely distributed in nature and usually found in decomposing organic matter or soil [1]. The species most frequently found in human infections are *Rhizopus*, *Lichtheimia* (formerly known as *Absidia* or *Mycocladus*), and *Mucor* [2]. Mucormycosis is an invasive fungal infection that presents high mortality rates due to its aggressive pattern, difficulties in its diagnosis, and intrinsic high resistance to many antifungals used in clinical settings [3]. Some risk factors are associated with the establishment of mucormycosis, such as diabetes, chemotherapy, organ transplantation, hematological diseases, and high serum levels of iron [4]. Mucorales are considered opportunistic pathogens because they are frequently related to infections in immunocompromised patients, and although their incidence is low, cases have been increasing during the last few decades, especially in this group of patients [4]. The most common clinical manifestations are rhino-cerebral, pulmonary, cutaneous, gastrointestinal, and disseminated infections [5]. Angioinvasion is an important characteristic of invasive mucormycosis, which consists of penetration of the endothelium and the formation of thrombosis and tissue necrosis, facilitating the hematological dissemination of the fungi [1].

A concerning factor that contributes to the bad prognosis of mucormycosis is its intrinsic resistance to many antifungal drugs, such as amphotericin B, posaconazole, and isavuconazole. Interestingly, these are still the recommended options to treat mucormycosis in clinical settings, indicating that the development of alternative antifungal therapies is an urgent need [6]. In addition, many cases of Mucorales infections have been recently associated with COVID-19 as a post-COVID syndrome, which highlights the necessity of improving the clinical management of these mycoses [7,8,9,10].

In this context, miltefosine has been seen as a promising repurposing drug for the treatment of fungal infections. It is a phospholipid analog from the alkyl phosphocholine class that was first developed as an antitumor agent, but studies have already shown its activity against a variety of microorganisms, including *Leishmania* spp. and *Trypanosoma cruzi* [11,12,13]. Its antifungal activity has been demonstrated in dermatophytes, *Cryptococcus* spp., *Candida* spp., *Sporothrix* spp., *Paracoccidioides* spp., *Histoplasma capsulatum*, *Coccidioides posadasii*, *Rhizopus* spp., *Aspergillus* spp., *Fusarium* spp., and *Scedosporium* spp. [14,15,16,17,18,19,20,21,22].

Little is known about the mechanism of action of miltefosine, especially in Mucorales. Some studies have already shown that it affects ergosterol biosynthesis and increases plasma membrane permeability in dimorphic fungi and *Scedosporium* species [15,23]. In addition, it seems to directly interact with ergosterol and glucosylceramide on fungal membranes, which could help to explain the disruption to the plasma membrane caused by miltefosine [23,24]. Since the exact mechanisms of action of miltefosine have not been clarified yet, the present study aims to evaluate the effects caused by this drug on different species of Mucorales. The null hypothesis was that miltefosine did not show significant difference in terms of cell alterations between different Mucorales species tested. In addition, miltefosine did not exhibit fungicidal activity in concentrations of up to 64 µg/mL against the Mucorales species, as was observed in several other fungi.

## 2. Materials and Methods

### 2.1. Microorganisms

*Rhizopus oryzae* UCP1295, *Rhizopus stolonifera* UCP1300, and *Rhizopus microsporus* var. *microsporus* UCP1304 isolated from the Brazilian Caatinga area were supplied by Galba Maria de Campos-Takaki from the Culture Collection (RENNORFUN - Rede Norte Nordeste de Fungos Filamentosos) from the Catholic University of Pernambuco, Recife, Brazil. Fungal stocks were kept in potato dextrose medium. Clinical strains of *Mucor velutinosus* H136BO and *Cunninghamella* spp. B926 were supplied by Marcio Nucci from the Mycology Laboratory of the University Hospital, Universidade Federal do Rio de Janeiro. To obtain conidia, cells were grown on potato dextrose agar plates for seven days at room temperature. Conidia were obtained by washing the plate surface with phosphate-buffered saline (PBS, pH 7.2), and hyphal fragments and debris were removed by filtration through a cell strainer (Falcon). The suspension was then centrifuged and cells were counted in Neubauer’s chamber to be used in the experiments. These five Mucorales species were used in all experiments of this study.

### 2.2. Antifungal Susceptibility Testing

The susceptibility of Mucorales species to miltefosine was determined according to EUCAST protocols, with modifications [25], using the broth microdilution method. A stock solution of miltefosine (Cayman Chemical Co., Ann Arbor, MI, USA) was prepared in dimethyl sulphoxide:ethanol (1:1) and maintained at −20 °C. To evaluate the minimum inhibitory concentration, miltefosine was serially diluted (0.062–64 μg/mL) and added to 96-well plates in the presence of a standardized suspension of conidia (3 × 10^4^), followed by an incubation in RPMI 1640 medium (Sigma-Aldrich, St. Louis, MO, USA). After 72 h at 37 °C in 5% CO_2_, cell growth was analyzed by OD readings at 600 nm in a spectrophotometer (Bio-Rad, Hercules, CA, USA). Cell viability was assessed using the XTT-reduction assay. The minimum inhibitory concentration (MIC_70_) of miltefosine was defined as the lowest concentration that inhibits 70% of fungal growth.

After the MIC analyses, to determine the minimum fungicidal concentration (MFC), aliquots of 10 μL of each well were plated on potato dextrose agar and incubated at room temperature for 3 days.

### 2.3. Anti-Biofilm Assays

Miltefosine activity was evaluated against Mucorales mature biofilms. Firstly, biofilms were formed in 96-well plates for 24 h [26,27]. Briefly, a conidia suspension of each species (1 × 10^5^) was added to each well and incubated for 1.5 h at 37 °C in 5% CO_2_ for the adhesion stage. Further, the supernatant was removed to discard non-adherent cells, and fresh RPMI 1640 medium was added for biofilm formation for 24 h at 37 °C. After that, the medium was removed and miltefosine (¼ MIC_70_–8 MIC_70_) was added. Positive control consisted of biofilm formation in the absence of miltefosine. An additional incubation for 24 h at 37 °C in 5% CO_2_ was performed to evaluate miltefosine activity. Biofilms were analyzed by three parameters [26,27]: crystal violet for overall biomass, safranin for extracellular matrix, and XTT-reduction assay for metabolic activity.

### 2.4. Susceptibility to SDS and NaCl

To evaluate the susceptibility to sodium dodecyl sulfate (SDS) and NaCl, conidia of Mucorales species (1 × 10^5^) were grown in 96-well plates containing RPMI in the presence of miltefosine (½ MIC_70_ or MIC_70_) and co-incubated with either SDS (sub-inhibitory concentration of 45 µg/mL for *Rhizopus* species and *M. velutinosus* and 90 µg/mL for *Cunninghamella* spp.) or 1% NaCl. Positive control consisted of cells grown in the absence of miltefosine. After 72 h incubation at 37 °C in 5% CO_2_, the cell viability was measured by using the XTT-reduction assay and readings were captured using a spectrophotometer (Bio-Rad, Hercules, CA, USA) at 490 nm.

### 2.5. Susceptibility in the Presence of Exogenous Ergosterol and Glucosylceramide

Mucorales susceptibility to miltefosine in the presence of exogenous ergosterol (Sigma-Aldrich, St. Louis, MO, USA) or GlcCer (purified from *R. stolonifer* UCP1300) was determined according to EUCAST protocols, with modifications [24,25]. Conidia of Mucorales species (3 × 10^4^) were incubated in the presence of miltefosine in serial dilutions as described above in the antifungal susceptibility test section. Whereas fungi incubated only with miltefosine were used as a control of MIC_70_ values, cells were also grown with miltefosine using RPMI supplemented with purified ergosterol (50 and 100 μg/mL) and GlcCer (50 and 100 μg/mL) to check their susceptibility to miltefosine in the presence of these lipids [23]. After 72 h incubation at 37 °C in 5% CO_2_, cell growth was measured in a spectrophotometer at 600 nm (Bio-Rad, Hercules, CA, USA) and cell viability was determined by the XTT-reduction assay, with the MIC of miltefosine defined as the lowest concentration that inhibits 70% of fungal growth.

### 2.6. Fluorescent Staining to Evaluate Alterations in the Fungal Cell

Calcofluor White (Sigma-Aldrich, St. Louis, MO, USA), concanavalin A (Sigma-Aldrich, St. Louis, MO, USA), Nile red (Sigma-Aldrich, St. Louis, MO, USA), 2′,7′-dichlorofluorescein diacetate (DCFH-DA) (Sigma-Aldrich, St. Louis, MO, USA), and JC-1 (ThermoFisher, Waltham, MA, USA) were used as fluorescent probes to evaluate alterations in cell wall sugars, membrane lipids, ROS production, and mitochondrial membrane potential, respectively. Fungal conidia (1 × 10^5^) were grown for 48 h at 37 °C in 5% CO_2_ in the presence of miltefosine (½ MIC_70_), and untreated cells were used as control. After a washing step, cells were stained with these fluorescent probes for 1 h at 37 °C in the dark. Then, the samples were washed three times to remove residual dye and 1 × 10^5^ cells were suspended in PBS. The fluorescence intensity was measured using the SpectraMax 340 microplate reader (Molecular Devices, San Jose, CA, USA) at the following wavelengths: Calcofluor White at 350 nm (excitation) and 432 nm (emission), concanavalin A at 495 nm (excitation) and 520 nm (emission), Nile Red at 550 nm (excitation) and 550 nm (emission), DCFH-DA at 492 nm (excitation) and 517 nm (emission), JC-1 at 475 nm (excitation) and 529 nm (green fluorescence), and 590 nm (red fluorescence) for the calculation of the red/green fluorescence intensity.

### 2.7. Transmission Electron Microscopy

The cells were centrifuged, washed with sterile PBS buffer, and processed for TEM via the following steps: (i) fixation in 2.5% glutaraldehyde, 4% formaldehyde, and 0.01 M calcium chloride in 0.1 M cacodylate buffer for 2 h at room temperature; (ii) three washes in 0.1 M cacodylate buffer; (iii) post-fixation with 1% buffered osmium tetroxide and 0.8% potassium ferrocyanide for 2 h at room temperature; (iv) three washes in 0.1 M cacodylate buffer; (v) sequential dehydration in acetone at 30%, 50%, 70% + 2.5%, 90%, and 100% uranyl acetate 3×; and (vi) embedded in Spurr resin. Ultrathin sections were prepared using an EM UC6 microtome (Leica Microsystems, Wetzlar, Germany), recovered on 300-mesh copper grids, stained with uranyl acetate and lead citrate, and observed on an FEI Morgagni transmission electron microscope (FEI Company, Hillsboro, OR, USA) operated at 80 kV.

### 2.8. Scanning Electron Microscopy

Mucorales species were grown in RPMI in the presence of ½ MIC_70_ of miltefosine with orbital agitation (150 rpm) for 72 h. Positive control consisted of cells grown in the absence of miltefosine. Mycelium samples were gently collected and processed via the following steps: (i) fixation in 2.5% glutaraldehyde and 4% formaldehyde in 0.1 M cacodylate buffer for at least 24 h at 4 °C, (ii) post-fixation in 1% osmium tetroxide in 0.1 M cacodylate buffer containing 1.25% potassium ferrocyanide and 5 mM CaCl_2_ for 30 min, (iii) dehydration in a graded ethanol series (30–100%), (iv) critical point drying in CO_2_, (v) adhesion of samples on aluminum stubs with a carbon tape, and (vi) coating with gold. Images were obtained in the scanning electron microscopes FEI Quanta 250 (FEI Company, Hillsboro, OR, USA) and Zeiss EVO 10 (Zeiss Company, Oberkochen, Germany). Images were processed using Photoshop software (version 24.6.0, Adobe, San Jose, CA, USA).

### 2.9. Antifungal Drug Synergy Assay

Synergistic interactions were detected by the checkerboard method according to EUCAST guidelines [25]. Conidia of Mucorales species (3 × 10^4^) were grown in 96-well plates containing RPMI in the presence of miltefosine (0.125–8 µg/mL) combined with posaconazole (0.5–64 µg/mL) or amphotericin B (0.78–50 µg/mL). After 72 h incubation at 37 °C in 5% CO_2_, the MIC_70_ was determined by readings at 600 nm and cell viability was assessed by XTT-reduction assay at 490 nm. Interactions were determined by two different methods: the fractional inhibitory concentration index (FICI) and the Bliss independence model.

The fractional inhibitory concentration index was calculated using the following formula: (MIC_70_ combined/MIC_70_ drug A alone) + (MIC_70_ combined/MIC_70_ drug B alone). The results were classified as synergistic effect, FICI of ≤0.5; no effect, FICI of >0.5–4.0; and antagonistic effect, FICI of >4.0 [28].

The Bliss independence model was performed according to Meletiadis and colleagues and Zhao and colleagues [29,30]. The following formula was used to assess the drug interaction: Eexp = Ea + Eb − Ea × Eb, in which Eexp is the expected efficacy of drug combination, Ea is the efficacy of drug A (miltefosine), and Eb is the efficacy of drug B (amphotericin B and posaconazole). The results were classified as synergistic effect, Eobs > Eexp; indifference, Eobs = Eexp; and antagonistic effect, Eobs < Eexp.

### 2.10. Statistical Analyses

All experiments were performed in triplicate in three independent experimental sets. The experimental results are presented as the mean ± standard deviation (SD). Data were analyzed by nonparametric Kruskal–Wallis one-way analysis of variance to compare the differences among the groups (the group treated with miltefosine and the control group without miltefosine). The individual comparisons of the groups were performed using a Bonferroni post-test. Statistical analyses were performed using GraphPad Prism v5.00 for Windows (GraphPad Software, San Diego, CA, USA). The 90% or 95% confidence interval was determined in all experiments, considering *p* < 0.05 a statistically significant difference.

## 3. Results

### 3.1. Susceptibility of Mucorales Species to Miltefosine

The minimum inhibitory concentration of miltefosine varied from 2 µg/mL for *M. velutinosus* and 4 µg/mL for the three *Rhizopus* species to 8 µg/mL for *Cunninghamella* spp. (Table 1). The Mucorales species tested were more susceptible to posaconazole than amphotericin B, which were used as a control since these antifungal drugs are usually chosen in clinical settings for the treatment of Mucorales infections (Table 1). Although MIC values for posaconazole were lower than the MIC for amphotericin B, only amphotericin B showed a fungicidal effect against the tree *Rhizopus* species at 25 µg/mL (Table 1). *Cunninghamella* spp. was the species most resistant to miltefosine and amphotericin B but was susceptible to posaconazole. *M. velutinosus* was susceptible to miltefosine at 1 µg/mL, the lowest MIC values for miltefosine found in this study, and susceptible to posaconazole at 2 µg/mL, but it was resistant to amphotericin B at 25 µg/mL (Table 1). Despite being susceptible to miltefosine and posaconazole, all drugs tested showed a fungistatic effect for *M. velutinosus*. The analysis of minimum fungicidal concentration revealed that all species remained viable in the presence of miltefosine, suggesting that it demonstrates fungistatic activity (Table 1). However, even posaconazole, which is used in clinical settings, displayed only a fungistatic effect against the Mucorales samples used in the study (Table 1).

### 3.2. Effect of Miltefosine on Biofilms

The potential of miltefosine against fungal biofilms was also evaluated. Regarding the fungal biomass, miltefosine decreased at least 50% of biofilm growth at 4 MIC for *R. oryzae* and *R. stolonifer* and at 8 MIC for *R. microsporus*, *Cunninghamella* spp., and *M. velutinosus* (Figure 1A). Biofilm matrix was 50% reduced at 4 MIC for *R. stolonifer* and at 8 MIC for *R. oryzae*, *R. microsporus*, *Cunninghamella* spp., and *M. velutinosus* (Figure 1B). Miltefosine decreased at least 50% of biofilm viability at 4 MIC for *R. stolonifer*, *Cunninghamella* spp., and *M. velutinosus* and at 8 MIC for *R. oryzae* and *R. microsporus* (Figure 1C). These results suggest that miltefosine not only acts against planktonic cells but also displays antifungal activity against biofilms formed by Mucorales species.

### 3.3. Susceptibility of Miltefosine-Treated Cells to Membrane Stressors

In order to elucidate the modifications that miltefosine causes to the fungal cell surface, the susceptibility to membrane stressors such as SDS and NaCl was evaluated in the presence of miltefosine, and all compounds were used at sub-inhibitory concentrations (Figure 2). Except for *R. oryzae*, which showed no alteration in susceptibility, all species were more susceptible to SDS in the presence of miltefosine (Figure 2A). *R. oryzae*, *Cunninghamella* spp., and *M. velutinosus* were more susceptible to NaCl in the presence of miltefosine (Figure 2B). However, the susceptibility to NaCl was not altered for *R. stolonifer* and *R. microsporus* in the presence of miltefosine (Figure 2B).

### 3.4. Susceptibility to Miltefosine in the Presence of Exogenous GlcCer and Ergosterol

To check whether miltefosine MIC_70_ values are altered in the presence of exogenous GlcCer and ergosterol, these molecules were added to the media and an antifungal susceptibility test was performed to compare MIC_70_ values of miltefosine alone and in the presence of GlcCer and ergosterol.

In the presence of GlcCer, MIC_70_ for miltefosine increased fourfold for *R. oryzae*, eightfold for *R. microsporus*, fourfold for *R. stolonifer*, eightfold for *Cunninghamella* spp., and twofold for *M. velutinosus* (Table 2). Regarding the addition of ergosterol, MIC_70_ for miltefosine increased at least fourfold for all species when ergosterol was added (Table 2).

### 3.5. Cell Alterations Caused by Miltefosine

In order to characterize the alterations in Mucorales cells caused by miltefosine, different fluorescent staining procedures were performed. Calcofluor white and concanavalin A were used as cell wall markers due to their ability to evaluate modifications in chitin and mannose residues, respectively. Treatment with miltefosine resulted in a reduction in chitin content for *Cunninghamella* spp. and *M. velutinosus* (Figure 3A), whereas a decrease in mannose residues was observed for all five fungi (Figure 3B).

Oxidative stress and depolarization of mitochondrial membranes were also evaluated using DCFH-DA and JC-1 staining, respectively. Miltefosine caused oxidative stress in *R. oryzae*, *R. stolonifer*, and *Cunninghamella* spp. since ROS staining increased compared to the untreated control (Figure 3C). Regarding the mitochondrial membrane, miltefosine led to a decreased depolarization in *R. microsporus*, *R. stolonifer*, and *Cunninghamella* spp. because the JC-1 ratio was reduced compared to untreated control (Figure 3D). Miltefosine treatment reduced the neutral lipid content for all Mucorales species tested since Nile Red staining intensity decreased compared to the untreated control (Figure 3E).

### 3.6. Transmission Electron Microscopy of Fungal Cells Treated with Miltefosine

Transmission electron microscopy (TEM) was performed to analyze whether morphological and ultrastructural changes are observed when fungal cells are treated with miltefosine ½ MIC_70_ values (Figure 4). Mucorales species were grown for 72 h in the absence of miltefosine and exhibited granules with different electron densities and some vacuoles with different sizes (Figure 4A,E,I,M,Q). In conditions treated for 72 h, cytoplasmic extravasation was observed in all cases (Figure 4B–D,F–H,J–L,N–P,R–T).

### 3.7. Scanning Electron Microscopy of Fungal Cells Treated with Miltefosine

Scanning electron microscopy (SEM) was performed to analyze whether miltefosine induced morphological alterations even at subinhibitory concentrations (miltefosine ½ MIC_70_ values) (Figure 5). Mucorales species growing for 72 h in the absence of miltefosine showed a mycelium containing non-septate hyphae, with spores germinated into hyphae (Figure 5A,C,E,G,I). When cells were treated with miltefosine, it was possible to observe an increase in the presence of rupture (Figure 5B) and amorphous cells (arrows in Figure 5D,F,H). The main alteration observed in *M. velutinosus* was the presence of more spores with a small germ tube and fewer developed hyphae (Figure 5H).

### 3.8. Evaluation of Synergism with Antifungal Drugs

The interaction of miltefosine with posaconazole and amphotericin B was evaluated. According to the FICI, miltefosine presented no effect with posaconazole, which was observed for all five species tested (Table 3). Although FICI did not indicate any positive interaction, a reduction in MIC_70_ values was seen for miltefosine and posaconazole in *R. stolonifer* and *Cunninghamella* spp. (Table 3). Regarding the interaction with amphotericin B, miltefosine showed a synergic effect only for *Cunninghamella* spp. (Table 3).

When the interaction was analyzed by the Bliss method, which considers the percentage of efficacy, miltefosine presented synergistic interaction with posaconazole for *R. oryzae*, *R. stolonifer*, *Cunninghamella* spp., and *M. velutinosus* and antagonistic interaction for *R. microsporus* (Table 4). Miltefosine showed synergistic interaction with amphotericin B for all species tested except for *M. velutinosus*, which presented an indeterminate effect (Table 4).

## 4. Discussion

Mucormycosis is a life-threatening infection that is difficult to treat since the causing agents affect mainly immunocompromised patients and are known to be intrinsically resistant to several antifungal drugs, including most azoles [31]. In this context, new options for mucormycosis treatment are needed, and drug repurposing has emerged as an interesting approach.

Miltefosine is a drug approved by the US Food and Drug Administration with an established antiparasitic activity, as it is used to treat visceral and cutaneous leishmaniasis, especially in countries such as India, Colombia, and Brazil [32]. In the last few years, several studies have reported the in vitro efficacy of miltefosine against fungal species, including yeasts and filamentous and dimorphic fungi, indicating broad-spectrum antifungal activity [19,22]. In this work, we describe the antifungal activity of miltefosine against mucormycosis-causing agents, a group of non-septate fungi all belonging to the Mucorales order. Our results show that MIC_70_ values for Mucorales species ranged from 2 to 8 µg/mL, corroborating a previous report in *Rhizopus* spp. that showed MICs from 2 to 16 µg/mL [22]. Unlike observations made with several other fungi, such as *Scedosporium* spp., *Sporothrix* spp., *Cryptococcus* spp., *Paracoccidioides* spp., *C. albicans*, and *F. oxysporum*, miltefosine did not exhibit fungicidal activity in concentrations up to 64 µg/mL against the Mucorales species used in this work [14,18,23,24,33]. This indicates a different manner of antifungal activity against non-septate fungi.

Castro and colleagues (2017) conducted an open-label clinical trial to assess the pharmacokinetic activity of miltefosine (Impavido) at a nominal dose of 2.5 mg/kg/day for 28 days for the oral treatment of cutaneous leishmaniasis [34]. According to the study, the median maximum concentration (Cmax) of miltefosine in plasma observed in children (n = 30) and adults (n = 29) was 22.7 μg/mL and 31.9 μg/mL, respectively [34]. The in vitro MIC_70_ values of miltefosine reported in our study were lower than the Cmax observed by Castro and colleagues in 2017 [34]. If we simply compare concentrations, the dosing regimen for the treatment of cutaneous leishmaniasis may result in a Cmax that would be sufficient for antifungal activity of miltefosine against *R. oryzae*, *R. microsporus*, *R. stolonifer*, *Cunninghamella* spp., and *M. velutinosus*, since Cmax > MIC_70_. However, a more refined pharmacokinetic/pharmacodynamic (PK/PD) relationship is necessary to better predict the efficacy of an antifungal drug and, consequently, enable the definition of the dosing regimen. Antifungal drugs can exhibit either concentration-dependent or time-dependent activity [35]. This dose–response relationship can be analyzed through PK/PD indexes, such as the Cmax in relation to the MIC (Cmax/MIC; a concentration-dependent measure), the area under the drug concentration curve in relation to MIC (AUC/MIC; a combination of both concentration and time measure), or the fraction of the interval in which the free drug concentration is above the MIC (fT > MIC; a time-dependent measure) [35,36,37]. As an example, we can mention liposomal amphotericin B, for which the Cmax/MIC is reported as the best PK/PD index to predict clinical response, and it may be ≥4.5 [38]. To identify the best PK/PD index that can predict the efficacy of miltefosine for the treatment of mucormycosis and, thus, design therapeutic dosing regimen, in vivo studies in infections models, for example, will be necessary.

Fungal biofilms are related to increased antifungal resistance and immune system evasion. In this context, mature biofilms are difficult to eradicate, and their presence on different surfaces may lead to new forms of contamination and worse prognosis during infection [39]. Miltefosine also exhibited anti-biofilm activity against Mucorales species, reducing biomass, extracellular matrix, and the viability of preformed biofilms at 4 or 8 MICs, which represents concentrations varying from 8 to 64 µg/mL. These results further establish the anti-biofilm activity of miltefosine since this drug is effective against biofilms of *C. albicans*, *F. oxysporum*, *Scedosporium* spp., and *Sporothrix* spp. [20,23,33,40].

As an alkylphospholipid, miltefosine acts by interacting with membrane lipids and altering membrane functions, as observed in studies using mimetic membranes and parasites. These studies observed that lipid rafts and sterol content play a major role in the interaction of miltefosine with plasma membranes [41,42]. Fungal lipid rafts are composed mainly of ergosterol and sphingolipids, which are important structures for cell division and morphogenesis [43]. In the present work, the addition of exogenous ergosterol or GlcCer (a major fungal sphingolipid) was able to significantly increase the MIC values of miltefosine in all Mucorales species. Previous works showed that exogenous ergosterol also increased miltefosine MIC in *Cryptococcus* spp. and *Candida krusei*, whereas GlcCer increased MIC in *Scedosporium aurantiacum* [23,24,44]. Subinhibitory concentrations of miltefosine were able to increase Mucorales species susceptibility to either SDS or NaCl (used as membrane stressors), indicating that miltefosine impacts the membrane integrity of these species, as has been reported for several other fungi [14,23,24]. These results highlight that part of miltefosine’s antifungal properties are related to an interaction with important fungal lipids that leads to an alteration in membrane integrity.

Cell alterations caused by miltefosine treatment were analyzed using fluorescent staining for different parameters. Neutral lipid content was reduced in all Mucorales species. It is known that miltefosine is able to interfere with lipid metabolism in parasites, as previous studies demonstrated that phosphatidylcholine concentration was decreased in *Trypanosoma cruzi* and *Leishmania donovani* [45,46]. Brilhante and colleagues demonstrated that ergosterol concentration was reduced after miltefosine treatment in *Histoplasma capsulatum* and *Coccidioides posadasii*, and a reduction in neutral lipid content was also observed in our previous work with *Scedosporium aurantiacum* [15,23].

Regarding cell wall parameters, chitin content was reduced only in *Cunninghamella* spp. and in *M. velutinosus*, whereas mannose residues were reduced in all five species. Taken together, these results suggest that miltefosine alters important fungal cell wall components. As observed in previous studies, *Cryptococcus* and *Sporothrix* yeasts had thinner cell walls after miltefosine treatment [14,24].

Oxidative stress and mitochondrial alterations have been described as additional mechanisms of action of miltefosine and are related to the induction of apoptosis, as demonstrated in other fungal species. In *Saccharomyces cerevisiae*, miltefosine induces cell death by interacting with COX9 (cytochrome c oxidase complex subunit of the electron transport chain), leading to a decrease in mitochondrial membrane potential [47]. Miltefosine also reduced mitochondrial membrane potential and increased intracellular ROS levels in *C. albicans*, *C. neoformans*, and *S. aurantiacum* [23,24,48]. In our study, the increase in ROS levels and decrease in mitochondrial membrane potential seemed to be species specific, with *Cunninghamella* spp. and *R. stolonifer* having both parameters altered, whereas only *M. velutinosus* had no alteration. Interestingly, even with these alterations being hallmarks of apoptosis and being observed in the Mucorales species used in this work, miltefosine had no fungicidal activity, as mentioned earlier. Xu and colleagues described that *R. oryzae* upregulated genes encoding heat shock proteins and oxidative stress response enzymes (superoxide dismutase and catalase) when intracellular ROS levels were increased [49]. The observations in our study led to hypotheses relating to possible stress response mechanisms that are activated in order to halt cell death processes induced by miltefosine.

The main observation in scanning electron microscopy (SEM) for all species treated with miltefosine was the presence of ruptured and amorphous cells, with the exception of *M. velutinosus*, which presented more spores with a small germ tube and fewer developed hyphae after miltefosine treatment. Previous reports have demonstrated some morphological alterations caused by classical antifungals in yeast and filamentous fungi. Dunyach and colleagues (2011) showed that the treatment of *Candida albicans* with caspofungin led to severe alterations in the cell wall and a loss of cell volume and cytoplasmic content, and consequently, there was no yeast–hypha transition [50]. Morphological alterations were also observed for *Sporothrix brasiliensis* treated with ketoconazole alone or in a complex with zinc, which included yeast–hyphae conversion, an increase in cell size, and cell wall damage to amorphous cells [51]. *Sporothrix schenckii* treated with amphotericin B led to an increase in single yeast and the appearance of amorphous cells compared to the control without treatment [52]. Few reports have demonstrated the morphological alterations in Mucorales species treated with classical antifungal drugs such as amphotericin B and posaconazole. Macedo and colleagues (2019) demonstrated that the combination of voriconazole with amphotericin B, posaconazole, or caspofungin led to small, rounded, and compact hyphal forms of *Rhizopus microsporus* compared to the growth control [53]. A recent study showed the morphological changes caused by selected compounds from the Pandemic Response Box^®^ library from Medicines for Malaria Venture (MMV), which mainly affected sporangia and spore formation from *Rhizopus oryzae* [54].

Drug combination is an important strategy to improve treatment and avoid the emergence of antifungal resistance. Due to the lack of clinical studies, combination therapy is not currently recommended as a first-choice treatment of mucormycosis [55]. Drogari-Apiranthitou and colleagues reported that double-drug combinations of amphotericin B, anidulafungin, and posaconazole were not synergic against *Rhizopus* spp. and *Lichthemia ramosa*. The only exception was the combination of amphotericin B with anidulafungin against *M. circinelloides* [56]. In the present work, we analyzed the efficacy of miltefosine combined with either amphotericin B or posaconazole via two distinct methods. FICI analysis revealed that the combination of miltefosine with amphotericin B was synergic against *Cunninghamella* spp., whereas it had no effect on the other species. However, it is important to note that, when combined with miltefosine, amphotericin B MIC values were reduced in all species except in *R. stolonifer*. The combination of miltefosine and posaconazole was also indifferent for all five species according to FICI. Using the Bliss independence method, miltefosine combined with amphotericin B was shown to act synergistically against all species except for *M. velutinosus*, for which the effect was indifferent. The combination of miltefosine and posaconazole had a synergic effect on all species with the exception of *R. microsporus*, for which the effect was antagonistic. The determination of a synergic effect between two drugs is highly dependent on several aspects, including the species tested and the method of analysis. In vitro, in vivo, and clinical trials might result in contrasting data. For example, the combination of the iron chelator deferasirox with liposomal amphotericin B or micafungin resulted in synergic effects in a diabetic murine model of mucormycosis; however, higher mortality was observed in the combination therapy group during a clinical trial [57,58]. Considering that the Bliss method revealed synergic interactions and that by using the FICI method we observed reductions in amphotericin B MIC values, in vivo studies are very important to see whether these drug combinations would improve the outcomes in infection models. If successful, combined drug therapy would be important for using lower drug concentrations, for preventing adverse effects in patients, and for the emergence of antifungal resistance, as it has already been reported that *Mucor circinelloides* has genes for efflux pumps responsible for azole resistance [59]. Although miltefosine is a drug that induces several cellular alterations and possibly has multiple molecular targets, further studies are needed to clarify the efficacy of combination treatment with the drugs of choice for mucormycosis treatment.

In spite of the results observed, this study has some limitations, such as the elucidation of the chemical interactions between miltefosine with ergosterol and glucosylceramide and the ability of miltefosine to alter the lipid composition of treated fungal cells, as observed for MDCK [60,61] and HepG2 cells [62], *Leishmania donovani* [46], and *Trypanosoma cruzi* [45]. The main limitation of the present study is the lack of an in vivo study to demonstrate the antifungal efficacy of miltefosine in Mucorales.

In conclusion, in the present work, we describe that miltefosine has fungistatic and anti-biofilm activity against five different mucormycosis etiologic agents and that this drug interacts with fungal lipids, destabilizing the plasma membrane and affecting the cell wall. Therefore, miltefosine could be considered for drug repurposing to improve the efficacy of mucormycosis treatment (Figure 6).

## Figures and Tables

**Figure 1 jof-09-01166-f001:**
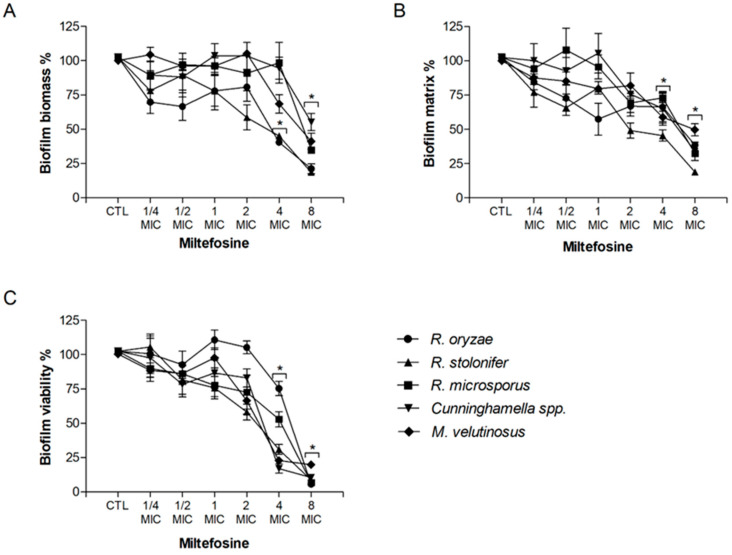
Effect of miltefosine on preformed biofilms of *R. oryzae* UCP1295, *R. stolonifer* UCP1300, *R. microsporus* UCP1304, *Cunninghamella* spp. B926, and *M. velutinosus* H136BO. Fungal biofilm was firstly formed in RPMI medium on polystyrene surface for 24 h and was then treated with different concentrations of miltefosine (¼–8 MIC) for another 24 h incubation. Intact fungal biofilms were considered controls (CTL, 100% biofilm) and their degradation due to treatment was compared to the control. Fungal biomass (**A**), extracellular matrix (**B**), and viability (**C**) were measured using violet crystal, safranin, and XTT-reduction assay, respectively. * *p* < 0.05, compared to zero (absence of drug) for each species.

**Figure 2 jof-09-01166-f002:**
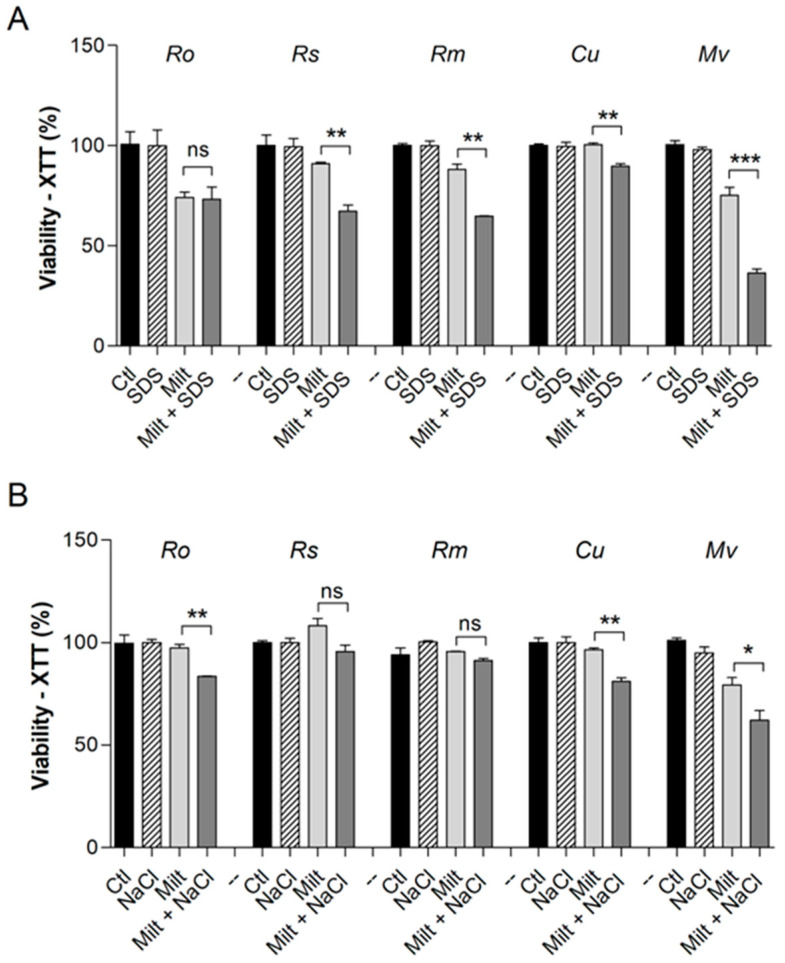
Susceptibility of Mucorales species to Miltefosine in the presence of surface stressors. SDS (90 µg/mL for *Cunninghamella* spp. and 45 µg/mL for the other species) was used as a membrane stressor (**A**). NaCl 1% was used as an osmotic stressor (**B**). The control represents fungal viability in the absence of stressors and miltefosine. Ro: *R. oryzae*; Rs: *R. stolonifer*; Rm: *R. microspores*; Cu: *Cunninghamella* spp.; Mv: *M. velutinosus*; Ctl: control; Milt: miltefosine; SDS: sodium dodecyl sulphate. * *p* < 0.05, ** *p* < 0.01, *** *p* < 0.001. ns—not significant. Experiments were performed in duplicate in three independent experimental sets.

**Figure 3 jof-09-01166-f003:**
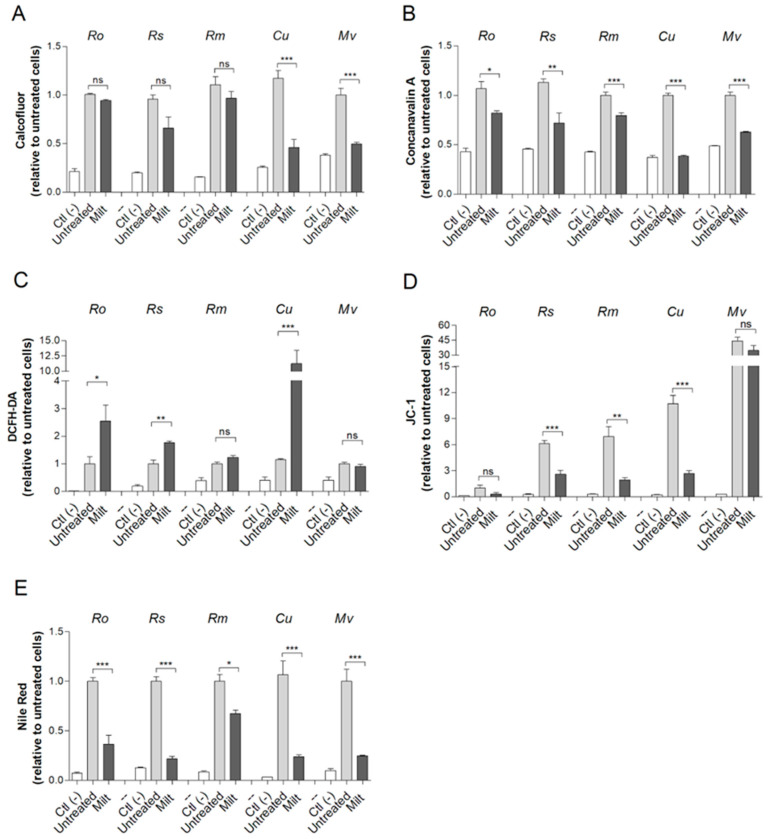
The effect of miltefosine on Mucorales species analyzed by fluorescent staining. Cells were grown in the presence of ½ MIC_70_ for 72 h at 37 °C. Chitin content was analyzed using calcofluor white (**A**). Concanavalin A was used to evaluate mannose residues (**B**). Oxidative stress; ROS was measured using DCFH-DA (**C**). The mitochondrial membrane polarization was measured using JC-1 (**D**). Neutral lipids were quantified using Nile Red stain (**E**). Ctl (–), a negative control that represents cells in the absence of fluorescent stain. Untreated, a positive control that represents cells stained with fluorescent stain but without drug treatment. Ro: *R. oryzae*; Rs: *R. stolonifer*; Rm: *R. microspores*; Cu: *Cunninghamella* spp.; Mv: *M. velutinosus*; Milt: miltefosine; ns: not significant. * *p* < 0.05; ** *p* < 0.01; *** *p* < 0.001. Experiments were performed in quadruplicate in three independent experimental sets.

**Figure 4 jof-09-01166-f004:**
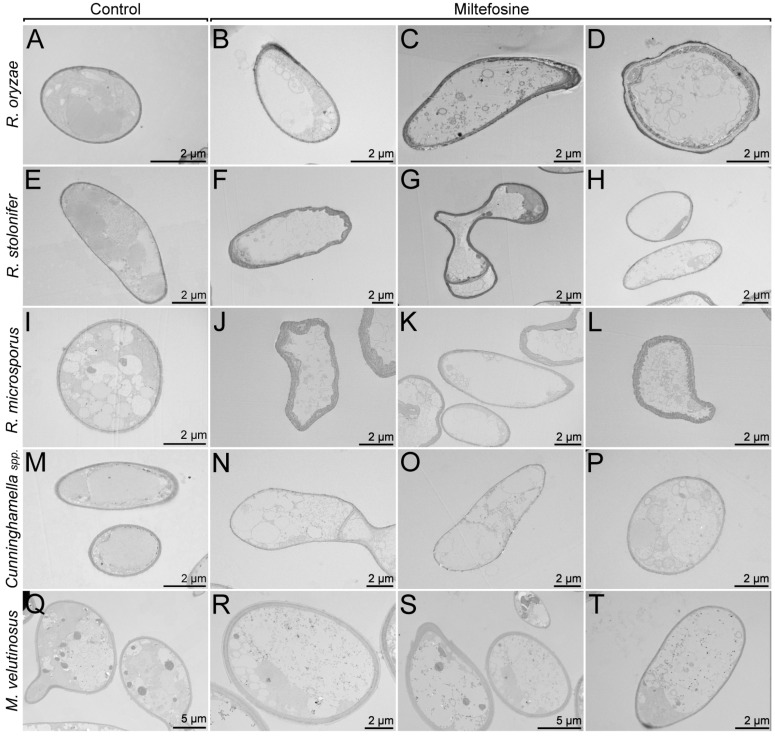
Mucorales alterations after exposure to miltefosine, evaluated by transmission electron microscopy. The control cultures (**A**,**E**,**I**,**M**,**Q**) showed granules with different electron densities and some vacuoles with different sizes. The treated samples (**B**–**D**,**F**–**H**,**J**–**L**,**N**–**P**,**R**–**T**) presented cytoplasmic extravasation.

**Figure 5 jof-09-01166-f005:**
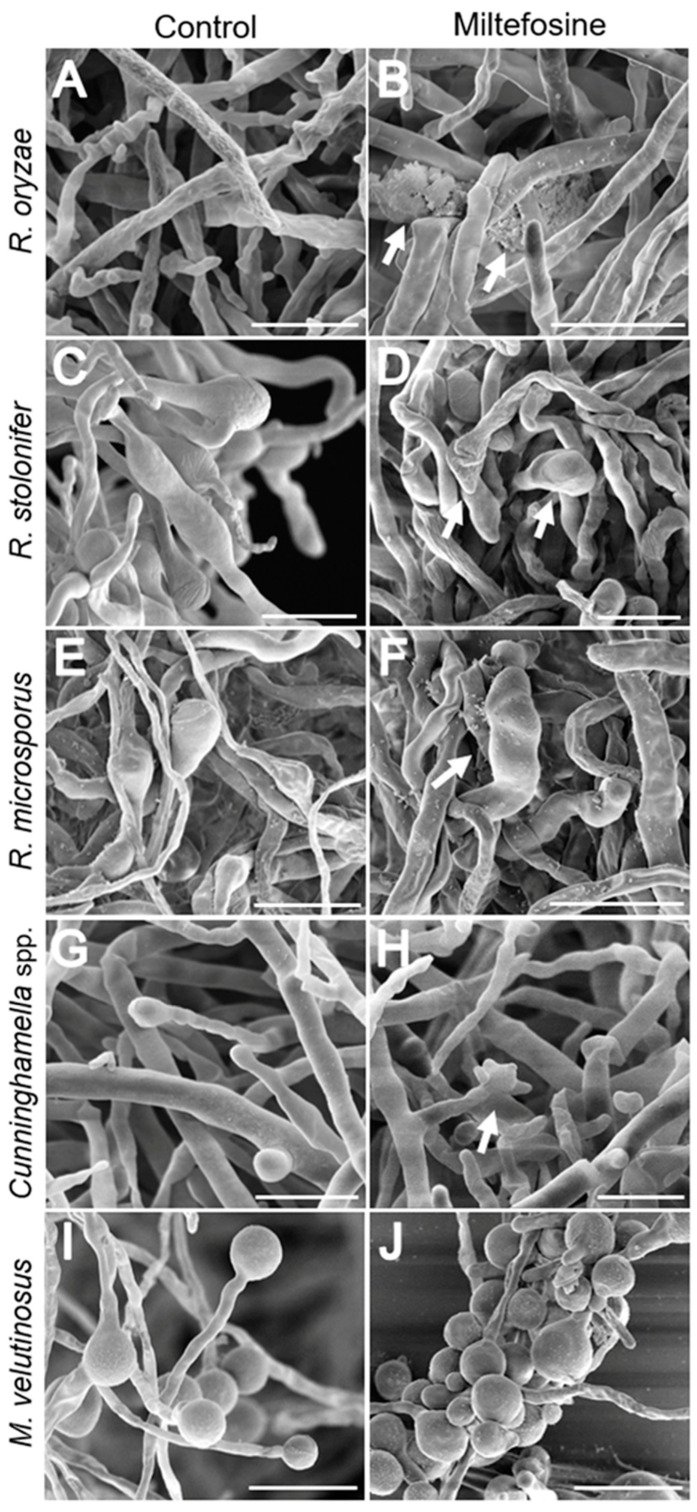
Mucorales alterations after exposure to miltefosine, evaluated by scanning electron microscopy. The control cultures (**A**,**C**,**E**,**G**,**I**) exhibited non-septate hyphae and treated samples (**B**,**D**,**F**,**H**,**J**) showed rupture and amorphous cells (arrows). Bar: 20 μm.

**Figure 6 jof-09-01166-f006:**
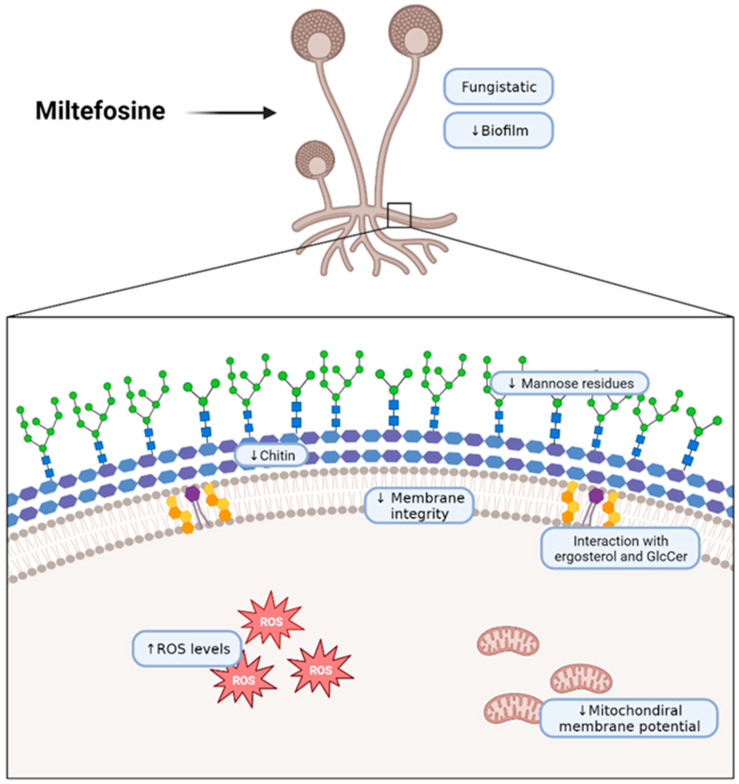
Schematic representation of the effects of miltefosine on fungal cells of the Mucorales species used in this work. The scheme wascreated with BioRender (https://app.biorender.com/illustrations/62b618fecae28690cdc1e912?slideId=b07dc37f-9866-421f-a4e4-3a17cc10ab37, accessed on 3 October 2023).

**Table 1 jof-09-01166-t001:** Minimum inhibitory and fungicidal concentration of miltefosine, amphotericin B, and posaconazole against *R. oryzae* UCP1295, *R. stolonifer* UCP1300, *R. microspores* UCP1304, *Cunninghamella* sp. B926, and *M. velutinosus* H136BO.

Species	Miltefosine		Amphotericin B		Posaconazole	
	MIC_70_	MFC	MIC_70_	MFC	MIC_70_	MFC
*R. oryzae*	4 µg/mL	>64 µg/mL	25 µg/mL	25 µg/mL	8 µg/mL	>16 µg/mL
*R. stolonifer*	4 µg/mL	>64 µg/mL	25 µg/mL	25 µg/mL	8 µg/mL	>16 µg/mL
*R. microsporus*	4 µg/mL	>64 µg/mL	25 µg/mL	25 µg/mL	4 µg/mL	>16 µg/mL
*Cunninghamella* spp.	8 µg/mL	>64 µg/mL	>50 µg/mL	>50 µg/mL	1 µg/mL	>16 µg/mL
*M. velutinosus*	2 µg/mL	>64 µg/mL	25 µg/mL	>50 µg/mL	1 µg/mL	>16 µg/mL

MIC: minimal inhibitory concentration. MFC: minimum fungicidal concentration.

**Table 2 jof-09-01166-t002:** The minimum inhibitory concentration of miltefosine when *R. oryzae* UCP1295, *R. stolonifer* UCP1300, *R. microspores* UCP1304, *Cunninghamella* sp. B926, and *M. velutinosus* H136BO were grown in the presence of exogenous GlcCer or ergosterol.

Species			MIC		
	Miltefosine	Miltefosine + GlcCer	Miltefosine + GlcCer	Miltefosine + Ergosterol	Miltefosine + Ergosterol
		50 µg/mL	100 µg/mL	50 µg/mL	100 µg/mL
*R. oryzae*	4 µg/mL	16 µg/mL	16 µg/mL	16 µg/mL	32 µg/mL
*R. stolonifer*	4 µg/mL	16 µg/mL	16 µg/mL	16 µg/mL	32 µg/mL
*R. microsporus*	4 µg/mL	16 µg/mL	16 µg/mL	16 µg/mL	32 µg/mL
*Cunninghamella* spp.	8 µg/mL	32 µg/mL	32 µg/mL	32 µg/mL	32 µg/mL
*M. velutinosus*	2 µg/mL	2 µg/mL	4 µg/mL	8 µg/mL	16 µg/mL

MIC: minimal inhibitory concentration. GlcCer: glucosylceramide purified from *R. stolonifer* UCP1300.

**Table 3 jof-09-01166-t003:** Antifungal activity of miltefosine, posaconazole, and amphotericin B—alone and in combination according to the Fractional Inhibitory Concentration Index (FICI)—against *R. oryzae* UCP1295, *R. stolonifer* UCP1300, *R. microspores* UCP1304, *Cunninghamella* sp. B926, and *M. velutinosus* H136BO. MIC values were used to analyze the interaction between miltefosine with posaconazole or amphotericin B.

Species	MIC_70_ Alone (µg/mL)			MIC_70_ Combined		FICI	
	Milt	Ampho B	Posa	Ampho B/Milt	Posa/Milt	Ampho B + Milt	Posa + Milt
*R. oryzae*	4	25	8	12.5/0.25	8/4	0.56 (N)	2.0 (N)
*R. stolonifer*	4	25	8	25/4	2/2	2.0 (N)	0.75 (N)
*R. microsporus*	4	25	4	12.5/4	4/4	1.5 (N)	2.0 (N)
*Cunninghamella* spp.	8	100	1	2.5/0.25	0.25/4	0.056 (S)	0.75 (N)
*M. velutinosus*	2	25	1	6.25/1	1/2	0.75 (N)	2.0 (N)

Milt: miltefosine; Ampho B: amphotericin B; Posa: posaconazole; MIC: minimal inhibitory concentration; S: synergistic interaction; A: antagonist interaction; N: no effect.

**Table 4 jof-09-01166-t004:** Antifungal activity of miltefosine, amphotericin B, and posaconazole—alone and in combination according to the Bliss independence method—against *R. oryzae* UCP1295, *R. microspores* UCP1304, *R. stolonifer* UCP1300, *Cunninghamella* sp. B926, and *M. velutinosus* H136BO.

Species	Efficacy of Drugs Alone			Efficacy of Combined Drugs					
	(% of Inhibition—½ MIC_70_)			Ampho B			Posa		
	Milt	Ampho B	Posa	*E* _obs_	*E* _exp_	Δ*E*, %	*E* _obs_	*E* _exp_	Δ*E*, %
*R. oryzae*	29.9	45.1	43.3	72.5	61.5	11.0 (S)	91.2	46;4	44.8 (S)
*R. microsporus*	2.7	35.7	57.3	19.5	18.2	1.3 (S)	20.4	33.5	−13.1 (A)
*R. stolonifer*	44.3	41.7	64.2	81.2	67.6	13.6 (S)	93.7	71.6	22.1 (S)
*Cunninghamella* spp.	0.5	0.4	10.3	89.9	1.5	88.4 (S)	82.1	12.1	70.0 (S)
*M. velutinosus*	38.6	44.8	36	66.1	66.1	0 (I)	51.6	49.3	2.3 (S)

Milt: miltefosine; Ampho B: amphotericin B; Posa: posaconazole; MIC: minimal inhibitory concentration; I: indeterminate; S: synergistic interaction; A: antagonist interaction; *E*_obs_: efficacy observed in the analysis; *E*_exp_: efficacy expected according to Bliss calculation; Δ*E*: difference between *E*_obs_ and *E*_exp_ (interaction).

## Data Availability

Data are contained within the article.

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
