# Peer review of "Miltefosine: A Repurposing Drug against Mucorales Pathogens"

_jof, 2023, doi:10.3390/jof9121166_

Round 1
Reviewer 1 Report
Comments and Suggestions for Authors
This interesting pre-clinical work analyzes in detail the main microbiological bases that may justify a possible role of Miltefosine in the treatment of mucormycosis. They conclude that miltefosine has no fungicidal activity (up to MIC 64 ug/ml) but has fungistatic and anti-biofilm activity and it interacts with fungal lipids, destabilizing the plasma membrane and affecting the cell wall.
The study is highly relevant, since miltefosine has already demonstrated activity in other fungal infections. The authors explore various forms of antifungal activity against a variety of Mucorales species: antifungal susceptibility testing (by EUCAST protocols), anti-biofilm assays, susceptibility to SDS and NaCl, susceptibility in the presence of exogenous ergosterol, fluorescent staining to evaluate alterations in the fungal cell, transmission and scanning electron microscopy and antifungal drug synergy assays.
The methodology is refined and the presentation is brilliant. The tables and figures faithfully reflect what is expressed in the text and are adequate in number, extension and quality. The work lays a solid experimental base for further preclinical and clinical studies on the role of miltefosine in the management of mucormycosis.
Our considerations to the work are minor:
-line 208 says M. velutinosus was resistant to posaconazole (MIC>64ug/ml) but in table 1 it appears with MIC 1 ug/ml.
-¿What does a 50% reduction at 4 MIC mean (line 223)? for a MIC of 4 ug/ml or for a concentration 4 times above the MIC?.
-line 300-303, does it refer to figure 3 or figure 4?.
-in figure 5 it is expressed: “more spores with a small germ tube and fewer developed hyphae”, is this a pathogenicity data observed with other antifungals?.
-In Figure 2, the susceptibility of Mucorales in the presence of surface stressors, shouldn't there be differences between the control group and SDS or NaCl?
-The reduction of mannose, the decrease of mitochondrial membrane depolarization and neutral lipid content and the increase of oxidative stress with miltefosine can translate into a safety problem as an antifungal drug?.
-Other studies of antifungal susceptibility in Mucorales express MIC90. In this work activity is expressed by MIC70, can this complicate the comparison of antifungal activity in other works?.
-Based on MIC90 of 2-4 ug/ml for amphotericin B, other studies have suggested the use of high doses of amphotericin B for the treatment of mucormycosis. With the MICs proposed for miltefosine in this work, would high doses also be recommended, for example higher than those used in Leishamaniasis?
-Despite the fact that synergy studies with FICI do not show synergy, but it is observed when analyzing by Bliss methodology, given the fungistatic activity of miltefosine but not fungicidal, could a systematic use of miltefosine in combination with other antifungals, for example with amphotericin B, be proposal for clinical practice?
Author Response
Reviewer 1
This interesting pre-clinical work analyzes in detail the main microbiological bases that may justify a possible role of Miltefosine in the treatment of mucormycosis. They conclude that miltefosine has no fungicidal activity (up to MIC 64 µg/ml) but has fungistatic and anti-biofilm activity and it interacts with fungal lipids, destabilizing the plasma membrane and affecting the cell wall.
The study is highly relevant, since miltefosine has already demonstrated activity in other fungal infections. The authors explore various forms of antifungal activity against a variety of Mucorales species: antifungal susceptibility testing (by EUCAST protocols), anti-biofilm assays, susceptibility to SDS and NaCl, susceptibility in the presence of exogenous ergosterol, fluorescent staining to evaluate alterations in the fungal cell, transmission and scanning electron microscopy and antifungal drug synergy assays.
The methodology is refined and the presentation is brilliant. The tables and figures faithfully reflect what is expressed in the text and are adequate in number, extension and quality. The work lays a solid experimental base for further preclinical and clinical studies on the role of miltefosine in the management of mucormycosis.
Our considerations to the work are minor:
- Line 208 says M. velutinosuswas resistant to posaconazole (MIC>64ug/ml) but in table 1 it appears with MIC 1 ug/ml.
Answer: That was a mistake. The text was corrected (Lines 220 – 224):
“M. velutinosus was susceptible to miltefosine at 1 µg/ml, the lowest MIC value for miltefosine found in this study, and also susceptible to posaconazole at 2 µg/ml, but it was resistant to amphotericin B at 25 µg/ml (Table 1). Despite being susceptible to miltefosine and posaconazole, all drugs tested showed fungistatic effect for M. velutinosus.”
- What does a 50% reduction at 4 MIC mean (line 223)? for a MIC of 4 ug/ml or for a concentration 4 times above the MIC?
Answer: “50% reduction at 4MIC” means that the pre-formed biofilm decreased 50% at the concentration of 4 times above the MIC. The concentrations that reduced parameters of pre-formed biofilms were 16 µg/ml for Rhizopus species, 32 µg/ml for Cunninghamella spp. and 8 µg/ml for M. velutinosus.
- Line 300-303, does it refer to figure 3 or figure 4?
Answer: It refers to Figure 4. The figure number was corrected (Lines 322 and 324).
- In figure 5 it is expressed: “more spores with a small germ tube and fewer developed hyphae”, is this a pathogenicity data observed with other antifungals?
Answer: The main observation in scanning electron microscopy (SEM) for all species treated with miltefosine was the presence of ruptured and amorphous cells with exception of M. velutinosus presenting more spores with a small germ tube and fewer developed hyphae after miltefosine treatment. According to the cell alterations analyzed by fluorescent staining procedures, especially for M. velutinosus, miltefosine decreased the chitin, mannose and neutral lipid content, suggesting that the germination process could be affected, as observed by SEM for M. velutinosus. These results alone do not conclude that there is an alteration in pathogenicity, but rather in fungal growth. These morphological changes may result in a reduction in pathogenicity, but more studies are needed to explain these mechanisms.
The text was inserted in the Discussion section (Lines 489 – 509):
The main observation in scanning electron microscopy (SEM) for all species treated with miltefosine was the presence of ruptured and amorphous cells with exception of M. velutinosus presenting more spores with a small germ tube and fewer developed hyphae after miltefosine treatment. Previous reports have demonstrated some morphological alterations caused by classical antifungals in yeast and filamentous fungi. Dunyach and colleagues (2009) showed that the treatment of Candida albicans with caspofungin led to severe alterations on the cell wall, loss on cell volume and cytoplasmic content, and consequently there were not yeast-hypha transition (Dunyach et al, 2009). Morphological alterations were also observed for Sporothrix brasiliensis treated with ketoconazole alone or in a complex with zinc, which included yeast hyphae conversion, an increase in cell size, cell wall damage amorphous cells (de Azevedo-França et al, 2021). Sporothrix schenckii treated with amphotericin B led to an increase of single yeast, compared to the control without treatment, and the appearance of amorphous cells (Borba-Santos et al, 2021). Few reports have demonstrated the morphological alterations in Mucorales species treated with classical antifungal drugs such as amphotericin B and posaconazole. Macedo and colleagues (2019) demonstrated that the combination of voriconazole with amphotericin B, posaconazole or caspofungin led to small, rounded and compact hyphal forms of Rhizopus microsporus when compared to the growth control (Macedo et al, 2019). A recent study showed the morphological changes caused by selected compounds from the Pandemic Response Box® library from Medicines for Malaria Venture (MMV), that mainly affected sporangia and spore formation from Rhizopus oryzae (Xisto et al, 2023).
Dunyach et al, 2009 - Dunyach C, Drakulovski P, Bertout S, Jouvert S, Reynes J, Mallié M. Fungicidal activity and morphological alterations of Candida albicans induced by echinocandins: study of strains with reduced caspofungin susceptibility. Mycoses. 2011 Jul;54(4):e62-8. doi: 10.1111/j.1439-0507.2009.01834.x.
de Azevedo-França JA, Borba-Santos LP, de Almeida Pimentel G, Franco CHJ, Souza C, de Almeida Celestino J, de Menezes EF, Dos Santos NP, Vieira EG, Ferreira AMDC, de Souza W, Rozental S, Navarro M. Antifungal promising agents of zinc(II) and copper(II) derivatives based on azole drug. J Inorg Biochem. 2021 Jun;219:111401. doi: 10.1016/j.jinorgbio.2021.111401.
Borba-Santos LP, Nucci M, Ferreira-Pereira A, Rozental S. Anti-Sporothrix activity of ibuprofen combined with antifungal. Braz J Microbiol. 2021 Mar;52(1):101-106. doi: 10.1007/s42770-020-00327-9.
Macedo D, Leonardelli F, Dudiuk C, Vitale RG, Del Valle E, Giusiano G, Gamarra S, Garcia-Effron G. In Vitro and In Vivo Evaluation of Voriconazole-Containing Antifungal Combinations against Mucorales Using a Galleria mellonella Model of Mucormycosis. J Fungi (Basel). 2019 Jan 8;5(1):5. doi: 10.3390/jof5010005.
Xisto MIDDS, Rollin-Pinheiro R, de Castro-Almeida Y, Dos Santos-Freitas GMP, Rochetti VP, Borba-Santos LP, da Silva Fontes Y, Ferreira-Pereira A, Rozental S, Barreto-Bergter E. Promising Antifungal Molecules against Mucormycosis Agents Identified from Pandemic Response Box®: In Vitro and In Silico Analyses. J Fungi (Basel). 2023 Jan 31;9(2):187. doi: 10.3390/jof9020187.
- In Figure 2, the susceptibility of Mucorales in the presence of surface stressors, shouldn't there be differences between the control group and SDS or NaCl?
Answer: The susceptibility of Mucorales species to SDS and NaCl was performed before the test with miltefosine to determine a sub-inhibitory concentration for each species. Therefore, the sub-inhibitory concentration of SDS and NaCl used does not show a significant difference in viability compared to the control.
- The reduction of mannose, the decrease of mitochondrial membrane depolarization and neutral lipid content and the increase of oxidative stress with miltefosine can translate into a safety problem as an antifungal drug?
Answer: Although miltefosine causes cell alterations as reduction of cell wall mannose content, decrease of mitochondrial membrane depolarization and neutral lipid content, and increase of oxidative stress, it is a drug commonly used for the treatment of visceral and cutaneous leishmaniasis and it is approved for use by oral administration in many countries (Reimão et al, 2020; Coordenação-Geral de Vigilância de Zoonoses e Doenças de Transmissão Vetorial, 2020). Since 2021, miltefosine was included in the WHO Model List of Essential Medicines for treating leishmaniasis and recently, miltefosine was designated as an orphan drug by the Food and Drug Administration (FDA, USA) for the treatment of invasive candidiasis (WHO 2021; FDA 2021). The commonly side effect of miltefosine is in the gastrointestinal tract causing symptoms as anorexia, nausea, vomiting and diarrhea. Miltefosine can occasionally led to hepatic toxicity and nephrotoxicity (Monge-Maillo and López-Vélez, 2015).
Reimão JQ, Pita Pedro DP, Coelho AC. The preclinical discovery and development of oral miltefosine for the treatment of visceral leishmaniasis: a case history. Expert Opin Drug Discov 2020;15:647–58. doi:10.1080/17460441.2020.1743674.
Coordenação-Geral de Vigilância de Zoonoses e Doenças de Transmissão Vetorial. 2020. NOTA INFORMATIVA No 13/2020-CGZV/DEIDT/SVS/MS 1.Ministério da Saúde SEI 25000.093184/2020-26/pg. 16.
World Health Organization. World Health Organization Model List of Essential Medicines – 22nd List, 2021. Geneva: World Health Organization; Licence: CC BY-NC-SA 3.0 IGO.
FDA. Search Orphan Drug Designations and Approvals. 2021. https://www.accessdata.fda.gov/scripts/opdlisting/oopd/detailedIndex.cfm?cfgrid key=843921.
Monge-Maillo B, López-Vélez R. Miltefosine for visceral and cutaneous leishmaniasis: drug characteristics and evidence-based treatment recommendations. Clin Infect Dis. 2015 May 1;60(9):1398-404. doi: 10.1093/cid/civ004.
- Other studies of antifungal susceptibility in Mucorales express MIC90. In this work activity is expressed by MIC70, can this complicate the comparison of antifungal activity in other works?
Answer: When the susceptibility test was made, with different drug concentrations, the maximum inhibitory capacity for each drug was observed. The maximum inhibition observed was 70% for miltefosine against the five Mucorales strains. Other studies on antifungal susceptibility in Mucorales used MIC50, MIC80 or MIC100. By verifying the inhibition rates, we can observe if the drug has more potential against a specific species/strain.
Kachuei R, Khodavaisy S, Rezaie S, Sharifynia S. In vitro antifungal susceptibility of clinical species belonging to Aspergillus genus and Rhizopus oryzae. J Mycol Med. 2016 Mar;26(1):17-21. doi: 10.1016/j.mycmed.2015.12.002. Epub 2016 Feb 4. PMID: 26852191.
Widmer F, Wright LC, Obando D, Handke R, Ganendren R, Ellis DH, Sorrell TC. Hexadecylphosphocholine (miltefosine) has broad-spectrum fungicidal activity and is efficacious in a mouse model of cryptococcosis. Antimicrob Agents Chemother. 2006 Feb;50(2):414-21. doi: 10.1128/AAC.50.2.414-421.2006. PMID: 16436691; PMCID: PMC1366877.
de Lira Mota KS, de Oliveira Pereira F, de Oliveira WA, Lima IO, de Oliveira Lima E. Antifungal activity of Thymus vulgaris L. essential oil and its constituent phytochemicals against Rhizopus oryzae: interaction with ergosterol. Molecules. 2012 Dec 5;17(12):14418-33. doi: 10.3390/molecules171214418. PMID: 23519243; PMCID: PMC6268362.
- Based on MIC90of 2-4 ug/ml for amphotericin B, other studies have suggested the use of high doses of amphotericin B for the treatment of mucormycosis. With the MICs proposed for miltefosine in this work, would high doses also be recommended, for example higher than those used in Leishamaniasis?
Answer: With the results obtained in this work, it is still not possible to give a definitive answer whether high doses of miltefosine would be recommended for the treatment of mucormycosis.
- The text was inserted in the Discussion section (Lines 412 – 434):
Castro and colleagues (2017) conducted an open-label clinical trial to assess the pharmacokinetic of miltefosine (Impavido) at a nominal dose of 2.5 mg/kg/day for 28 days for the oral treatment of cutaneous leishmaniasis. According to the study, the median Cmax (maximum concentration) of miltefosine in plasma observed in children (n = 30) and adults (n = 29) were 22.7 μg/mL and 31.9 μg/mL, respectively [1]. The in vitro MIC70 values of miltefosine reported in our study were lower than the Cmax observed by Castro and colleagues (2017). If we simply compare concentrations, the dosing regimen for the treatment of cutaneous leishmaniasis may result in a Cmax that would be sufficient for antifungal activity of miltefosine against R. oryzae, R. microsporus, R. stolonifer, Cunninghamella sp. and M. velutinosus, since Cmax > MIC70. However, a more refined pharmacokinetic/pharmacodynamic (PK/PD) relationship is necessary to better predict the efficacy of an antifungal drug and, consequently, enable the definition of the “best” dosing regimen. Antifungal drugs can exhibit either concentration-dependent or time-dependent activity [5]. This dose-response relationship can be analyzed through PK/PD indexes, such as the Cmax in relation to the MIC (Cmax/MIC; a concentration-dependent measure), the area under the drug concentration curve in relation to MIC (AUC/MIC; a combination of both concentration and time measure) or the fraction of the interval in which the free drug concentration is above the MIC (fT > MIC; a time-dependent measure) [2, 3, 5]. As an example, we can mention the liposomal amphotericin B, which the Cmax/MIC is reported as the best PK/PD index to predict clinical response and it may be ≥ 4.5 [4]. To identify the best PK/PD index that can predict the efficacy of miltefosine for the treatment of mucormycosis and, thus, design therapeutic dosing regimen, in vivo studies in infections models, for example, will be necessary.
[1] Castro MD, Gomez MA, Kip AE, Cossio A, Ortiz E, Navas A, Dorlo TP, Saravia NG. Pharmacokinetics of Miltefosine in Children and Adults with Cutaneous Leishmaniasis. Antimicrob Agents Chemother. 2017 Feb 23;61(3):e02198-16. doi: 10.1128/AAC.02198-16.
[2] Lepak AJ, Andes DR. Antifungal pharmacokinetics and pharmacodynamics. Cold Spring Harb Perspect Med. 2014 Nov 10;5(5):a019653. doi: 10.1101/cshperspect.a019653.
[3] Gonzalez JM, Rodriguez CA, Agudelo M, Zuluaga AF, Vesga O. Antifungal pharmacodynamics: Latin America's perspective. Braz J Infect Dis. 2017 Jan-Feb;21(1):79-87. doi: 10.1016/j.bjid.2016.09.009.
[4] Gómez-López A. Antifungal therapeutic drug monitoring: focus on drugs without a clear recommendation. Clin Microbiol Infect. 2020 Nov;26(11):1481-1487. doi: 10.1016/j.cmi.2020.05.037.
[5] Howard A, Hope W. Assessment of Antifungal Pharmacodynamics. J Fungi (Basel). 2023 Feb 1;9(2):192. doi: 10.3390/jof9020192.
- Despite the fact that synergy studies with FICI do not show synergy, but it is observed when analyzing by Bliss methodology, given the fungistatic activity of miltefosine but not fungicidal, could a systematic use of miltefosine in combination with other antifungals, for example with amphotericin B, be proposal for clinical practice?
Answer: Considering that Bliss method revealed synergic interactions and that using FICI method we observed reductions in amphotericin B MIC values, in vivo studies are very important to see whether these drug combinations would improve the outcomes in infection models. If successful, combined drug therapy would be important to use lower drug concentrations, to prevent adverse effects in patients and the emergence of antifungal resistance as it was already reported that Mucor circinelloides has genes for efflux pumps responsible for azole resistance.
The text above was inserted in the manuscript (Lines 533 – 539).
Nagy G, Kiss S, Varghese R, Bauer K, Szebenyi C, Kocsubé S, Homa M, Bodai L, Zsindely N, Nagy G, Vágvölgyi C, Papp T. Characterization of Three Pleiotropic Drug Resistance Transporter Genes and Their Participation in the Azole Resistance of Mucor circinelloides. Front Cell Infect Microbiol. 2021 Apr 14;11:660347. doi: 10.3389/fcimb.2021.660347. PMID: 33937100; PMCID: PMC8079984.
Reviewer 2 Report
Comments and Suggestions for Authors
This research is under the scope of this journal; the topic is relevant for readers, and this research deals with potentially significant knowledge to the field.
However, there are some concerns about the present manuscript:
Introduction
“The most common clinical manifestations are rhino-cerebral, pulmonary, cutaneous, gastrointestinal, and disseminated infections” please consider: PMID: 37755045
Please re-write the null hypothesis in the aim section.
Material and Methods?
Number of samples in each experiment
- Sample size calculation is not clear. Please better describe the primary outcome utilized, standard deviation and the mean average among groups
Statistical Analysis?
There are many mistakes in the references section and in the text
The discussion is also misleading. What is the novelty of this paper???
Please add the limitations of the study in the discussion and provide an adequate debate. Discussion should be better organized.
Author Response
Reviewer 2
This research is under the scope of this journal; the topic is relevant for readers, and this research deals with potentially significant knowledge to the field.
However, there are some concerns about the present manuscript:
- Introduction:
“The most common clinical manifestations are rhino-cerebral, pulmonary, cutaneous, gastrointestinal, and disseminated infections” please consider: PMID: 37755045
Answer: The reference PMID:37755045 was added. (Line 44).
PMID:37755045 – “Mora-Martínez A, Murcia L, Rodríguez-Lozano FJ. Oral Manifestations of Mucormycosis: A Systematic Review. J Fungi (Basel). 2023 Sep 16;9(9):935. doi: 10.3390/jof9090935. PMID: 37755045; PMCID: PMC10533187.”
- Please re-write the null hypothesis in the aim section.
Answer: The null hypothesis was re-written (Lines 71 - 75):
“The null hypothesis was that miltefosine did not show significant difference in terms of cell alterations between different Mucorales species tested. In addition, miltefosine did not exhibit fungicidal activity in concentrations up to 64 µg/ml against the Mucorales species as it was observed several other fungi.”
- Material and Methods? Number of samples in each experiment.
Answer: The number of samples used in each experiment was five samples, one sample of each species: Rhizopus oryzae, Rhizopus stolonifer, Rhizopus microsporus var. microsporus, Mucor velutinosus and Cunninghamella spp.
The sentence was inserted in the manuscript in Material and Methods section (Line 89 – 90): “These five Mucorales species were used in all experiments done in this study.”
- Sample size calculation is not clear. Please better describe the primary outcome utilized, standard deviation and the mean average among groups. Statistical Analysis?
Answer: The sample size was one strain from each species used in this study. This is described in Material and Method section, in sub-section “2.1.Microorganisms”. For a better understanding, the sentence “These five Mucorales species were used in all experiments in this study” were inserted in this section (Line 89 – 90).
As suggested, the “Statistical analyses” section is rewritten (Lines 201 – 209): “All experiments were performed in triplicate, in three independent experimental sets. The experimental results are presented as the mean ± standard deviation (SD). Data were analyzed by nonparametric Kruskal–Wallis one-way analysis of variance to compare the differences among the groups (the group treated with miltefosine and the control group, without miltefosine). The individual comparisons of the groups were performed using a Bonferroni post-test. Statistical analyses were performed using GraphPad Prism v5.00 for Windows (GraphPad Software, San Diego, CA, USA). The 90% or 95% confidence interval was determined in all experiments, considering p < 0.05 a statistically significant difference.”
- There are many mistakes in the references section and in the text
Answer: - The mistakes in the reference section were corrected, since the file of EndNote style was changed for the MDPI EndNote style in the manuscript.
- Letter “E” was added in Figure 4, because it was missing (Lines 327 – 345).
- The Figure number was corrected (Lines 322 and 324).
- spp. was corrected in Tables 1, 3 and 4 (Cunninghamella spp.); and in Lines
480 and 519.
- “in vitro” was corrected to italic formatting (Line 400).
- A reference was inserted in Line 406.
- The funding information of Borba-Santos, L.P. was corrected (Line 575).
- The discussion is also misleading. What is the novelty of this paper???
Answer: Many studies have reported the in vitro antifungal effect of miltefosine in several fungal species. Previous studies that included Mucorales species in their analyses were focused on the determination of MIC values of miltefosine or synergic potential with other drugs. Mucormycosis agents are non-septate fungi and this reflects significant differences in growth and cell biology when compared to other fungi. Therefore, studies on the effects of a non-classical antifungal drug in different species of fungi are important to report possible differences (that might impact the use of this drug in the future) or similarities (that could contribute to assess the antifungal activity spectrum of the drug). For example, in our study we report that miltefosine has a fungistatic activity against the strains used as opposed to the majority of the studies in other fungal species that reported fungicidal activity. Also, in our study we showed the antibiofilm properties as well as several alterations in Mucorales species cells caused by miltefosine treatment. These results could contribute to the understanding of miltefosine mechanism of action in these pathogens.
Widmer F, Wright LC, Obando D, Handke R, Ganendren R, Ellis DH, Sorrell TC. Hexadecylphosphocholine (miltefosine) has broad-spectrum fungicidal activity and is efficacious in a mouse model of cryptococcosis. Antimicrob Agents Chemother. 2006 Feb;50(2):414-21. doi: 10.1128/AAC.50.2.414-421.2006. PMID: 16436691; PMCID: PMC1366877.
Biswas C, Sorrell TC, Djordjevic JT, Zuo X, Jolliffe KA, Chen SC. In vitro activity of miltefosine as a single agent and in combination with voriconazole or posaconazole against uncommon filamentous fungal pathogens. J Antimicrob Chemother. 2013 Dec;68(12):2842-6. doi: 10.1093/jac/dkt282. Epub 2013 Jul 16. PMID: 23861311.
- Please add the limitations of the study in the discussion and provide an adequate debate.
Answer: This study demonstrates the miltefosine effect on five Mucorales species through different tests. The miltefosine showed the fungistatic effect on all species tested and also effect on mature biofilms, in addition to presenting interaction with fungal lipids since sub-inhibitory concentrations of miltefosine were able to increase Mucorales species susceptibility to either SDS or NaCl (used as membrane stressors) and the susceptibility in the presence of exogenous ergosterol or glucosylceramide. Alterations in fungal cell as reduction of cell wall mannose content, the decrease of mitochondrial membrane depolarization and neutral lipid content, and the increase of oxidative stress, were also observed in the Mucorales treated with miltefosine. All these results indicate that miltefosine has multiple molecular targets.
This paragraph was inserted in de Discussion section (Lines 543 – 548): “In spite of the results observed, this study has some limitations as the elucidation of the chemical interactions between miltefosine with ergosterol and glucosylceramide, the ability of miltefosine to alter the lipid composition of treated fungal cells as observed for MDCK [3, 4] and HepG2 cells [5], Leishmania donovani [1] and Trypanosoma cruzi [2]. The main limitation of the present study is the lack of an in vivo study to demonstrate the antifungal efficacy of miltefosine in Mucorales.”
[1] Rakotomanga M, Blanc S, Gaudin K, Chaminade P, Loiseau PM. Miltefosine affects lipid metabolism in Leishmania donovani promastigotes. Antimicrob Agents Chemother. 2007 Apr;51(4):1425-30. doi: 10.1128/AAC.01123-06.
[2] Lira R, Contreras LM, Rita RM, Urbina JA. Mechanism of action of anti-proliferative lysophospholipid analogues against the protozoan parasite Trypanosoma cruzi: potentiation of in vitro activity by the sterol biosynthesis inhibitor ketoconazole. J Antimicrob Chemother. 2001 May;47(5):537-46. doi: 10.1093/jac/47.5.537.
[3] Haase R, Wieder T, Geilen CC, Reutter W. The phospholipid analogue hexadecylphosphocholine inhibits phosphatidylcholine biosynthesis in Madin-Darby canine kidney cells. FEBS Lett. 1991 Aug 19;288(1-2):129-32. doi: 10.1016/0014-5793(91)81018-4
[4] Wieder T, Haase A, Geilen CC, Orfanos CE. The effect of two synthetic phospholipids on cell proliferation and phosphatidylcholine biosynthesis in Madin-Darby canine kidney cells. Lipids. 1995 May;30(5):389-93. doi: 10.1007/BF02536296. Erratum in: Lipids 1995 Jul;30(7):681. PMID: 7637558.
[5] Jiménez-López JM, Carrasco MP, Segovia JL, Marco C. Hexadecylphosphocholine inhibits phosphatidylcholine biosynthesis and the proliferation of HepG2 cells. Eur J Biochem. 2002 Sep;269(18):4649-55. doi: 10.1046/j.1432-1033.2002.03169.x.
- Discussion should be better organized.
Answer: We agree with the referee’s comment. In the revised version, we hope that the discussion is now better organized.
- The text was inserted in the Discussion section (Lines 412 – 434):
Castro and colleagues (2017) conducted an open-label clinical trial to assess the pharmacokinetic of miltefosine (Impavido) at a nominal dose of 2.5 mg/kg/day for 28 days for the oral treatment of cutaneous leishmaniasis. According to the study, the median Cmax (maximum concentration) of miltefosine in plasma observed in children (n = 30) and adults (n = 29) were 22.7 μg/mL and 31.9 μg/mL, respectively [1]. The in vitro MIC70 values of miltefosine reported in our study were lower than the Cmax observed by Castro and colleagues (2017). If we simply compare concentrations, the dosing regimen for the treatment of cutaneous leishmaniasis may result in a Cmax that would be sufficient for antifungal activity of miltefosine against R. oryzae, R. microsporus, R. stolonifer, Cunninghamella sp. and M. velutinosus, since Cmax > MIC70. However, a more refined pharmacokinetic/pharmacodynamic (PK/PD) relationship is necessary to better predict the efficacy of an antifungal drug and, consequently, enable the definition of the “best” dosing regimen. Antifungal drugs can exhibit either concentration-dependent or time-dependent activity [5]. This dose-response relationship can be analyzed through PK/PD indexes, such as the Cmax in relation to the MIC (Cmax/MIC; a concentration-dependent measure), the area under the drug concentration curve in relation to MIC (AUC/MIC; a combination of both concentration and time measure) or the fraction of the interval in which the free drug concentration is above the MIC (fT > MIC; a time-dependent measure) [2, 3, 5]. As an example, we can mention the liposomal amphotericin B, which the Cmax/MIC is reported as the best PK/PD index to predict clinical response and it may be ≥ 4.5 [4]. To identify the best PK/PD index that can predict the efficacy of miltefosine for the treatment of mucormycosis and, thus, design therapeutic dosing regimen, in vivo studies in infections models, for example, will be necessary.
[1] Castro MD, Gomez MA, Kip AE, Cossio A, Ortiz E, Navas A, Dorlo TP, Saravia NG. Pharmacokinetics of Miltefosine in Children and Adults with Cutaneous Leishmaniasis. Antimicrob Agents Chemother. 2017 Feb 23;61(3):e02198-16. doi: 10.1128/AAC.02198-16.
[2] Lepak AJ, Andes DR. Antifungal pharmacokinetics and pharmacodynamics. Cold Spring Harb Perspect Med. 2014 Nov 10;5(5):a019653. doi: 10.1101/cshperspect.a019653.
[3] Gonzalez JM, Rodriguez CA, Agudelo M, Zuluaga AF, Vesga O. Antifungal pharmacodynamics: Latin America's perspective. Braz J Infect Dis. 2017 Jan-Feb;21(1):79-87. doi: 10.1016/j.bjid.2016.09.009.
[4] Gómez-López A. Antifungal therapeutic drug monitoring: focus on drugs without a clear recommendation. Clin Microbiol Infect. 2020 Nov;26(11):1481-1487. doi: 10.1016/j.cmi.2020.05.037.
[5] Howard A, Hope W. Assessment of Antifungal Pharmacodynamics. J Fungi (Basel). 2023 Feb 1;9(2):192. doi: 10.3390/jof9020192.
- The text was inserted in the Discussion section (lines 489 – 509):
The main observation in scanning electron microscopy (SEM) for all species treated with miltefosine was the presence of rupture and amorphous cells with exception of M. velutinosus presenting more spores with a small germ tube and fewer developed hyphae after miltefosine treatment. Previous reports have demonstrated some morphological alterations caused by classical antifungals in yeast and filamentous fungi. Dunyach and colleagues (2009) showed that the treatment of Candida albicans with caspofungin led to severe alterations on the cell wall, loss on cell volume and cytoplasmic content, and consequently there were not yeast-hypha transition (Dunyach et al, 2009). Morphological alterations were also observed for Sporothrix brasiliensis treated with ketoconazole alone or in a complex with zinc, which included yeast hyphae conversion, an increase in cell size, cell wall damage amorphous cells (de Azevedo-França et al, 2021). Sporothrix schenckii treated with amphotericin B led to an increase of single yeast, compared to the control without treatment, and the appearance of amorphous cells (Borba-Santos et al, 2021). Few reports have demonstrated the morphological alterations in Mucorales species treated with classical antifungal drugs such as amphotericin B and posaconazole. Macedo and colleagues (2019) demonstrated that the combination of voriconazole with amphotericin B, posaconazole or caspofungin led to small, rounded and compact hyphal forms of Rhizopus microsporus when compared to the growth control (Macedo et al, 2019). A recent study showed the morphological changes caused by selected compounds from the Pandemic Response Box® library from Medicines for Malaria Venture (MMV), that mainly affected sporangia and spore formation from Rhizopus oryzae (Xisto et al, 2023).
Dunyach et al, 2009 - Dunyach C, Drakulovski P, Bertout S, Jouvert S, Reynes J, Mallié M. Fungicidal activity and morphological alterations of Candida albicans induced by echinocandins: study of strains with reduced caspofungin susceptibility. Mycoses. 2011 Jul;54(4):e62-8. doi: 10.1111/j.1439-0507.2009.01834.x.
de Azevedo-França JA, Borba-Santos LP, de Almeida Pimentel G, Franco CHJ, Souza C, de Almeida Celestino J, de Menezes EF, Dos Santos NP, Vieira EG, Ferreira AMDC, de Souza W, Rozental S, Navarro M. Antifungal promising agents of zinc(II) and copper(II) derivatives based on azole drug. J Inorg Biochem. 2021 Jun;219:111401. doi: 10.1016/j.jinorgbio.2021.111401.
Borba-Santos LP, Nucci M, Ferreira-Pereira A, Rozental S. Anti-Sporothrix activity of ibuprofen combined with antifungal. Braz J Microbiol. 2021 Mar;52(1):101-106. doi: 10.1007/s42770-020-00327-9.
Macedo D, Leonardelli F, Dudiuk C, Vitale RG, Del Valle E, Giusiano G, Gamarra S, Garcia-Effron G. In Vitro and In Vivo Evaluation of Voriconazole-Containing Antifungal Combinations against Mucorales Using a Galleria mellonella Model of Mucormycosis. J Fungi (Basel). 2019 Jan 8;5(1):5. doi: 10.3390/jof5010005.
Xisto MIDDS, Rollin-Pinheiro R, de Castro-Almeida Y, Dos Santos-Freitas GMP, Rochetti VP, Borba-Santos LP, da Silva Fontes Y, Ferreira-Pereira A, Rozental S, Barreto-Bergter E. Promising Antifungal Molecules against Mucormycosis Agents Identified from Pandemic Response Box®: In Vitro and In Silico Analyses. J Fungi (Basel). 2023 Jan 31;9(2):187. doi: 10.3390/jof9020187.
- The text was inserted in the Discussion section (lines 533 – 539):
Considering that Bliss method revealed synergic interactions and that using FICI method we observed reductions in amphotericin B MIC values, in vivo studies are very important to see whether these drug combinations would improve the outcomes in infection models. If successful, combined drug therapy would be important to use lower drug concentrations, to prevent adverse effects in patients and the emergence of antifungal resistance as it was already reported that Mucor circinelloides has genes for efflux pumps responsible for azole resistance (Nagy et al, 2021).
Nagy G, Kiss S, Varghese R, Bauer K, Szebenyi C, Kocsubé S, Homa M, Bodai L, Zsindely N, Nagy G, Vágvölgyi C, Papp T. Characterization of Three Pleiotropic Drug Resistance Transporter Genes and Their Participation in the Azole Resistance of Mucor circinelloides. Front Cell Infect Microbiol. 2021 Apr 14;11:660347. doi: 10.3389/fcimb.2021.660347. PMID: 33937100; PMCID: PMC8079984